# Multi-Modal Adaptive Fusion Transformer Network for the Estimation of Depression Level

**DOI:** 10.3390/s21144764

**Published:** 2021-07-12

**Authors:** Hao Sun, Jiaqing Liu, Shurong Chai, Zhaolin Qiu, Lanfen Lin, Xinyin Huang, Yenwei Chen

**Affiliations:** 1School of Software Technology, Zhejiang University, Hangzhou 315048, China; sunhaoxx@zju.edu.cn; 2College of Information Science and Engineering, Ritsumeikan University, Kusatsushi 5250058, Shiga, Japan; gr0302kv@ed.ritsumei.ac.jp (J.L.); is0538kr@ed.ritsumei.ac.jp (S.C.); chen@is.ritsumei.ac.jp (Y.C.); 3College of Computer Science and Technology, Zhejiang University, Hangzhou 315048, China; qiuzhaolin@zju.edu.cn; 4School of Education, Soochow University, Suzhou 215006, China; huangxinyin@suda.edu.cn; 5Research Center for Healthcare Data Science, Zhejiang Lab, Hangzhou 311121, China

**Keywords:** depression detection, transformer, multi-modal, audio-visual, adaptive fusion, multi-task, regression, classification, PHQ-8 score

## Abstract

Depression is a severe psychological condition that affects millions of people worldwide. As depression has received more attention in recent years, it has become imperative to develop automatic methods for detecting depression. Although numerous machine learning methods have been proposed for estimating the levels of depression via audio, visual, and audiovisual emotion sensing, several challenges still exist. For example, it is difficult to extract long-term temporal context information from long sequences of audio and visual data, and it is also difficult to select and fuse useful multi-modal information or features effectively. In addition, how to include other information or tasks to enhance the estimation accuracy is also one of the challenges. In this study, we propose a multi-modal adaptive fusion transformer network for estimating the levels of depression. Transformer-based models have achieved state-of-the-art performance in language understanding and sequence modeling. Thus, the proposed transformer-based network is utilized to extract long-term temporal context information from uni-modal audio and visual data in our work. This is the first transformer-based approach for depression detection. We also propose an adaptive fusion method for adaptively fusing useful multi-modal features. Furthermore, inspired by current multi-task learning work, we also incorporate an auxiliary task (depression classification) to enhance the main task of depression level regression (estimation). The effectiveness of the proposed method has been validated on a public dataset (AVEC 2019 Detecting Depression with AI Sub-challenge) in terms of the PHQ-8 scores. Experimental results indicate that the proposed method achieves better performance compared with currently state-of-the-art methods. Our proposed method achieves a concordance correlation coefficient (CCC) of 0.733 on AVEC 2019 which is 6.2% higher than the accuracy (CCC = 0.696) of the state-of-the-art method.

## 1. Introduction

Depression is one of the most severe mental disorders worldwide, which can severely affect people’s daily lives. There are still many debates about the causes of depression, but the three main causes are psychological, genetic, and social and environmental problems. As numerous people suffer from depression worldwide, it has become important to detect depression using computer-aided methods. The detection of depression is a challenging task as many of its symptoms are widely distributed in ages, gender, regions, and cultures. The limitations in the detection of depression using computer-aided methods necessitate the development of more effective methods to accomplish this task. In recent years, numerous approaches have been proposed to automatically detect the levels of depression.

Recently, many computer-aided methods, especially deep learning (DL)-based methods, have been proposed to estimate depression levels. Concretely, DL models take data collected from people suffering from depression as input and then output predicted results. These data collected from patients include videos, audio, texts, and other information, which can be perceived as different modalities. Although many contributions have been made in the field of depression detection, many unresolved issues still exist. First, the data used in detecting depression are generally rather long in temporal dimension as depression cannot be detected in a short time. However, processing long-term sequences has always been a challenge in DL [1]. Recurrent neural networks (RNNs) have been widely used for the extraction of temporal information to detect depression in the past few years [2,3,4], including the baseline model provided by the Detecting Depression with AI Sub-challenge (DDS) [5] which is one of the challenges in the AVEC 2019 [5]. Traditional RNN structures, including long short-term memory (LSTM), bi-directional LSTM (Bi-LSTM), and gated recurrent units (GRUs), can effectively process short-term time sequences. However, they cannot satisfactorily process long-term sequences. As the length of sequences increases, the performance of RNNs rapidly decreases due to the forgetting problem. The forgetting issue of RNNs means that RNNs will lose the primary information when reading later series at the scenario of processing long-term sequences. Second, there have always been various fusion approaches to fuse information from different modalities, early fusion and late fusion for example. It has been proven that multi-modal learning can improve the performance of depression detection in recent research [2,6,7]. However, many proposed DL-based methods train their models in a simple multi-modal manner, ignoring the differences in the contribution of different modalities.

In this study, we have proposed a multi-modal adaptive fusion transformer network to address these two challenges mentioned above. For the first challenge, a transformer model [8] has been proposed to process long-term sequences while effectively handling the forgetting issue. To the best of our knowledge, this is the first application of the transformer method for depression detection. For the second challenge, it is important to weight modalities as they have different contributions to the final results. Thus, we proposed an adaptive late-fusion strategy to fuse results from different modalities adaptively. In the Adaptive Late-Fusion strategy, we will increase the weights of effective modalities or features, while lessening the weights of ineffective modalities or features. Identifying which modalities or features are effective is also one of purposes of our research, which will be shown in Section 5.3. In addition, we employ a multi-task representation learning strategy in our work as many current research studies [3,9] have proven that multi-task learning can positively contribute to depression detection. Therefore, in this study, we apply multi-modal learning and multi-task learning to the transformer-based network to detect depression on the AVEC 2019 DDS Challenge dataset [5].

The remainder of this paper is organized as follows. Section 2 summarizes the related works. Section 3 describes the details of the proposed network. Section 4 presents experiments with corresponding results in Section 5. Section 6 presents the discussions. Finally, Section 7 concludes the paper.

## 2. Related Work

### 2.1. Depression Detection Using Single-Modal Information

As data used in DL for depression detection are time series, irrespective of how many modalities there are, it is important to effectively extract temporal information from every single modality. Currently, the most commonly used methods for extracting temporal information for a single modality are RNN models, including LSTM and GRUs. For example, the baseline model of the AVEC 2019 DDS Challenge [5] used a single GRU layer to process time series to detect depression levels. A hierarchical Bi-LSTM was used in [2] to extract temporal information in order to obtain information with different temporal scales. Qureshi, S.A. used the traditional LSTM structure in [3] to obtain sequential features for every single modality to estimate the levels of depression. Although RNN families are widely used for extracting temporal information, they still have some drawbacks, the most significant being the problem called *Forgetting*. The forgetting issue is explained as an RNN model that loses previous information when processing long-term sequences. Although LSTM and GRUs have been proposed to mitigate the negative impact of the forgetting problem, unsatisfactory results are achieved while processing extremely long-term sequences. This forgetting issue limits the sequential length that RNN models can process. The forgetting issue can be handled better now that the transformer model [8] has been proposed. As a transformer model [8] has a pure attention structure, the impact of forgetting is small, allowing the model to process longer sequences than traditional RNN families.

While original transformer models have been successfully used in natural language processing tasks, recent research studies have employed transformer models in other fields, such as image processing and emotion recognition image. In particular, Gulati, A. fused a convolutional neural network (CNN) model and transformer model to process images, which has been called conformer in paper [10]. In the field of emotion recognition, authors in [11] first used a transformer model to predict emotions. Because of the similarity between emotion recognition and depression detection, numerous research studies [12] have applied emotion methodologies to depression detection. In this work, we used a transformer model to predict the levels of depression; to the best of our knowledge, this is the first time a transformer model is used in this field.

### 2.2. Depression Detection Using Multi-Modal Information

Multi-modal learning is one of the most important strategies in depression detection. As the data to be analyzed in depression detection are composed of several modalities, such as video, audio, and text, it is relatively common to perform multi-modal learning. Currently, numerous research studies [2,6] have proven that multi-modal learning can improve the accuracy and robustness of depression level prediction. The most commonly used modalities include audio, videos, and texts, which are collected through interviews with patients suffering from depression, with their corresponding features, such as MFCCs and AUposes. For example, the AVEC 2019 DDS Challenge [5] dataset includes features extracted from original audios and videos, such as MFCC, eGeMAPS, and AUposes.

The multi-modal fusion strategy can be roughly divided into early fusion and late-fusion. Early fusion means fusion of data at the feature level, whereas late-fusion means fusion of data at the decision level. Nowadays, most methods fuse information in the early fusion stage. For instance, authors in [13] used the bag-of-words model to encode audio and visual features and then fused them to perform multi-modal learning for depression detection. Rodrigues Makiuchi, M. in [14] used texts generated from the original speech audio by Google Cloud’s speech recognition service with their hidden embedding extracted from pretrained BERT [15] model while concatenating all modalities, achieving a concordance correlation coefficient (CCC) score of 0.69 on the AVEC 2019 DDS Challenge dataset. Aside from audio, video, and text modalities, methods proposed in [16] employed body gestures as one of the modalities to perform early fusion. For late-fusion, the most representative method is the baseline model of the AVEC 2019 DDS Challenge [5], which first obtains results from each uni-modality and then takes the average as the final prediction.

However, most of the current methods did not explicitly weigh modalities with different performances, whether using early or late-fusion. In our work, we propose an adaptive late-fusion strategy that can leverage the importance of different modalities. Specifically, we weigh modalities according to their performances, which means that we assign high weights to modalities with high performance and low weights to those with poor performance to obtain final late-fusion results. According to our experimental results, we can infer that the proposed Adaptive Late-Fusion can improve the performance of depression detection.

## 3. Proposed Technologies

In this section, we first show an overall description of our multi-modal adaptive fusion transformer network and then provide a detailed description of the Transformer Encoder module, encoding the time series data from each modality. Subsequently, we elaborate on how our multi-task methods use a multi-task representation learning network for PHQ-8 regression and 5-class classification. Finally, we fuse acoustic and visual modalities in Adaptive Late-Fusion to conduct the final depression level prediction. The architecture of the proposed method is presented in Figure 1.

To illustrate the effectiveness of the transformer model in the depression detection, we employ the Transformer Encoder to extract temporal information. After features from every modality are processed by the *Transformer Encoder* and the *Embedding FC Block* presented in Figure 1, they are fed to two *FC Blocks* designed for multi-task learning, which will be later described in more detail in the Section 3.3.

### 3.1. Input Stream

In the AVEC 2019 DDS Challenge dataset, two main modalities can be obtained, namely, audio and video modalities. Each modality type contains several kinds of features, such as MFCC from audio and AUposes from video, which can be obtained by the methods provided by the AVEC 2019 DDS Challenge. For every type of feature obtained from the AVEC 2019 DDS Challenge dataset, the model is independently trained. The results obtained from each type of feature are fused in the independent stage called Adaptive Late-Fusion. For every type of feature, the transformer model is designed to extract temporal information, and its detailed structure will be described in the next section, i.e., *Transformer Encoder*. Suppose pre-processed features have the shape of $R^{bs*t*d}$, where *bs* standard is the batch size, *t* standard is the temporal frames, and *d* standard is the feature dimension. After they have been processed by the Transformer Encoder, the features with the shape of $R^{bs*t*d}$ are averaged in the temporal *t* dimension to obtain the shape of $R^{bs*d}$. The averaged features are fed to the *Embedding FC Block* to obtain features with the same dimension, which are treated as hidden embeddings representing every feature from each modality. Each *FC Block* consists of a rectified linear unit (ReLU) activation layer, a dropout layer, and a linear layer. The dropout layer in the *FC Block* is designed to overcome overfitting during training. The hidden embeddings are finally passed to the two FC Layer Blocks to perform predictions on two tasks: PHQ-8 regression and 5-class classification. After the results from each modality’s feature are obtained, we employ Adaptive Late-Fusion to obtain the final prediction results in terms of the PHQ-8 scores.

### 3.2. Transformer Encoder

The Transformer Encoder structure employed in this work has been detailed in Figure 2. Following [8], we use the naive Transformer Encoder structure, along with the Positional Encoding module, Multi-head Attention module, and Feed-Forward module in our work. In both the Multi-head Attention and Feed-Forward modules, data streams are designed as a shortcut structure with additive and normalization operations. An entire single Transformer Encoder layer architecture is repeated by *N* times to form a complete Transformer Encoder. Before being fed to the Transformer Encoder, input streams are processed by the Positional Encoding module to alter the positional information. Before being fed into the Multi-head Attention module, an input stream will be independently mapped to three sub-streams represented as *Q*, *K*, and *V*, respectively. Then, the Multi-head Attention module will perform global self-attention from *Q*, *K*, and *V*. If the head number of the Multi-head Attention module is greater than one, the Multi-head Attention module will perform the self-attention in different temporal scales. The Feed-Forward module is a simple feed-forward structure composed of two fully connected layers.

The Positional Encoding (PE) module is used to add positional information to the original input. The Positional Encoding model is important because the Transformer Encoder has a pure attention structure without any positional information. We use the same positional encoding method as [8], whose formula is shown in Equation (Equation 1):(1)$$\begin{array}{c} \mathrm{PE}\left(pos,2i\right)=sin\left(\frac{pos}{{10,000}^{\frac{2i}{d_{model}}}}\right)\\ \mathrm{PE}\left(pos,2i+1\right)=cos\left(\frac{pos}{{10,000}^{\frac{2i}{d_{model}}}}\right), \end{array}$$
where pos denotes the position (i.e., 0, 1, 2, ...) of every frame in features; *i* denotes the indices of elements in every single frame; and $d_{model}$ denotes the dimension of input features.

The Self-Attention module is designed to map a query (Q) and a set of key (K)-value (V) pairs to an attention value (Z). The Q, K, V are represented as Equation (Equation 2):(2)$$\begin{array}{c} Q=W_{Q}X\in R^{F\times 1}\\ K=W_{K}X\in R^{E\times 1}\\ V=W_{V}X\in R^{E\times 1}, \end{array}$$
where X represents the origin inputs, while Q, K, and V denote the query vector, key vector and value vector, respectively. Suppose the dimensions of Q, K, V are F,E,E, respectively. $W_{Q},W_{K},W_{V}$ are linear transform matrices for Q, K, V, respectively, which are learned to find best Q, K, V during the training process.

In self-attention, we first calculate the similarity between Q and K as Equation (Equation 3):
(3)$$\begin{array}{c} A=\frac{{\mathrm{QK}}^{T}}{\sqrt{E}}\in R^{F\times E}, \end{array}$$
where A is a similarity matrix or score matrix with a dimension of F×E. Its element $a_{ji}$ can be represented as:
(4)$$\begin{array}{c} a_{ji}=q_{j}k_{i}/\sqrt{E}, \end{array}$$
where $q_{j}$ and $k_{i}$ are *j*-th element of Q and *i*-th element of K, respectively. We use softmax to normalize $a_{ji}$ as:
(5)$$\begin{array}{c} a_{ji}^{{}^{\prime }}=softmax\left(a_{ji}\right)=exp\left(a_{ji}\right)/\sum_{k=1}^{E}exp\left(a_{jk}\right), \end{array}$$

The attention value ($z_{j}$) for $q_{j}$ can be represented as: (6)$$\begin{array}{c} z_{j}=\sum_{i=1}^{E}a_{ji}^{{}^{\prime }}v_{i}, \end{array}$$ where $v_{i}$ is the *i*-th element of V. The $Z\in R^{F\times 1}$ can be represented as:(7)$$\begin{array}{c} Z=A^{{}^{\prime }}V, \end{array}$$ where $A^{{}^{\prime }}$ is the normalized similarity matrix, whose element is $a_{ji}^{{}^{\prime }}$.

The final Feed-Forward module is made up of two fully-connected layers whose hidden units can be specified as hyperparameters.

### 3.3. Multi-Task Learning

Inspired by current multi-task learning works, we incorporate an auxiliary task (depression classification) to enhance the main task of depression level regression (estimation). To achieve the purpose of multi-task learning, after the features are processed by the *Embedding FC Block*, the hidden embedding for each type of feature will be fed to two *FC Blocks* to separately perform two tasks, i.e., PHQ-8 regression and 5-class classification. The *FC Blocks* comprise of an ReLU activation, a dropout layer, and a linear layer. Since we can only achieve the original PHQ-8 regression task using the AVEC 2019 DDS Challenge dataset, we generate 5-class classification labels from the original PHQ-8 score labels, as detailed in the *Data Processing* section.

Our multi-task loss function in the training stage can be formulated as:(8)$$\begin{array}{c} Loss=a*L_{re}+b*L_{cl}, \end{array}$$
where $L_{re}$ and $L_{cl}$ are loss functions for PHQ-8 regression and 5-class classification, respectively. *a* and *b* in Equation (Equation 8) are designed to leverage the coefficient between these two tasks and can be set as hyperparameters. Specifically, the loss function for PHQ-8 regression can be formulated as follows:(9)$$\begin{array}{c} L_{re}=1-\frac{2S_{\hat{y}y}}{S_{\hat{y}}^{2}+S_{y}^{2}+{\left(\bar{\hat{y}}-\bar{y}\right)}^{2}}, \end{array}$$
where $\hat{y}$ and *y* denote the predicted depression levels and true labels with $\bar{\hat{y}}$ and $\bar{y}$ denoting their corresponding mean values, $S_{\hat{y}}$ and $S_{y}$ denote the variances of $\hat{y}$ and *y* and $S_{\hat{y}y}$ means the covariance of them. We employ the commonly used cross-entropy loss as the loss function of our 5-class classification task, which is shown as follows:(10)$$\begin{array}{c} L_{cl}=-\frac{1}{N}\sum_{i=1}^{N}\sum_{c=1}^{C}1_{\left[c=y_{i}\right]}\log p_{i,c}, \end{array}$$
where *C* denotes the number of classification classes; *N*, the number of samples; $1_{\left[c=y_{i}\right]}$, a binary indicator; and $p_{i,c}$, the predicted probability that sample *i* belongs to class *c*.

### 3.4. Adaptive Late-Fusion

To fuse results from different modalities and adjust weights for each type of feature adaptively, we employ the proposed late-fusion strategies called Adaptive Late-Fusion to fuse results obtained from every single feature.

The general late-fusion strategy that is widely used takes the average from results obtained from each feature of different modalities, which can be formulaically expressed as follows:(11)$$FinalPredictions\_Averaged=\frac{\sum_{m=0}^{M}Predictions_{m}}{count(m)},$$
where *M* denotes the number of selected features, and $Predictions_{m}$ denotes the predictions from feature *m*. In this study, the general late-fusion strategy is known as Averaged Late-Fusion. The Adaptive Late-Fusion method proposed in our work aims to increase the weights of features with high performance while decreasing the weights of features with low performance. Specifically, we calculate the weights for each feature and take the weighted average from all modalities. Weights are calculated according to the CCC from each type of feature, and thus, the feature types with higher CCC will have larger weights in the proposed Adaptive Late-Fusion. The formulaic expression of our proposed Adaptive Late-Fusion is shown as follows:(12)$$\begin{array}{cc} {\mathrm{CCC}}_{Sum} & =\sum_{m=0}^{M}{\mathrm{CCC}}_{m}\\ FinalPredictions\_Weighted & =\sum_{m=0}^{M}\frac{\left(Predictions_{m}*{\mathrm{CCC}}_{m}\right)}{{\mathrm{CCC}}_{Sum}}, \end{array}$$
where *M* denotes the number of selected features; $Predictions_{m}$, the predictions from feature *m*; ${\mathrm{CCC}}_{Sum}$, the sum of CCCs for all features; and ${\mathrm{CCC}}_{m}$, the CCC score of feature *m*.

We implement our Adaptive Late-fusion Strategy in two ways. In one way, we select all modalities and all types of features and fuse the results from them, which means that the results obtained will account for every modality. In the other way, we only fuse the top *M* features ranked by the CCC [17] metric, which means that we will drop features with poor performance.

## 4. Experiments

### 4.1. The AVEC 2019 DDS Challenge Dataset

The DDS dataset was obtained from AVEC 2019 [5], where the level of depression (PHQ-8 questionnaire [18]) was assessed from audiovisual recordings of patients’ clinical interviews conducted by a virtual agent driven by a human as a Wizard-of-Oz (WoZ) [19]. The recording audio has been transcribed by Google Cloud’s speech recognition service and annotated for a variety of verbal and nonverbal features. Each interview in the AVEC 2019 DDS dataset comprises interview IDs, PHQ-8 binary labels, PHQ-8 scores, and the participant’s gender. The dataset contains baseline features extracted from audiovisual recordings by common frameworks based on open-source toolboxes. It spans three expressions levels: functional low-level descriptors (hand-crafted), bag-of-words, and unsupervised deep representations. The audio features are provided by openSMILE [20], and the video features are provided by openFace [21].

For every sample in the AVEC 2019 DDS dataset, the PHQ-8 scores range ∈[0,24]. Following [3], we define the cut-points at [0,5,10,15,20] for minimal depression, mild depression, moderate depression, moderately severe depression, and severe depression, respectively. The distribution of the AVEC 2019 DDS dataset is shown in Figure 3. The dataset includes MFCC, Bow_MFCC, eGeMAPS, Bow_eGeMAPS, DS_DNet, and DS_VGG for audio and FAUs, BoVW, ResNet, and VGG for video, where Bow indicates the bag-of-word method; DS, the deep spectrogram; and DNet and VGG, the data processed by pretrained DenseNet and VGG_Net, respectively. In this dataset, every modality feature has the shape of $R^{t*d}$, where *t* denotes the length of the sequence, and *d* represents the dimension of the modality.

### 4.2. Data Processing

Because the data sequences are too long to fit in 48GB of memory, which is our best GPU capacity with double RTX3090 GPU cards, we must shorten the dataset’s original data. To shorten the sequences, we sample *N* frames from the original features for every modality feature. Specifically, we evenly split the sequence into *s* segments; for each segment, we randomly sample *L = N/s* successive frames. Finally, we concatenate the *s* segments obtained from each segment. Consequently, we can obtain *N* frames from each kind of feature in this manner. For different types of features in the AVEC 2019 DDS dataset, we select different *N* and *s*, which can be treated as hyperparameters as the dimensions of different features are different.

We generate the MFCC_functional, eGeMAPS_functional, and AUpose_functional from MFCC, eGeMAPS, and AUpose, respectively, using the approach provided by the AVEC 2019 DDS to enhance the modality and avoid the side effect of processing extremely long-term sequences. Specifically, the functional features have the same lengths as 1768 and the mean value and standard deviation of the original data.

To achieve the goal of multi-task learning, we obtain classification labels from the original PHQ-8 scores, as illustrated by [3]. The corresponding relationships between the original PHQ-8 scores and 5-class classification labels and the label distributions are presented in Table 1.

### 4.3. Evaluation Functions

We use well-known standard evaluation metrics for depression detection to evaluate regression/classification results. We use the Concordance Correlation Coefficient (CCC) [17] as a measure of PHQ-8 estimated scores (regression task), which is the common metric in dimensional depression detection to measure the agreement between true PHQ-8 scores (*y*) and predicted PHQ-8 scores ($\hat{y}$). The CCC is formulated as follows:(13)$$\begin{array}{c} CCC=\frac{2S_{\hat{y}y}}{S_{\hat{y}}^{2}+S_{y}^{2}+{\left(\bar{\hat{y}}-\bar{y}\right)}^{2}}, \end{array}$$
where $S_{\hat{y}}$ and $S_{y}$ denote the variances of $\hat{y}$ and *y*, whereas $S_{\hat{y}y}$ denote the corresponding covariance value. The CCC is based on Lin’s calculation [17]. The range of the CCC is from −1 to 1, where −1 represents perfect disagreement and 1 represents perfect agreement.

As another measure for the regression task, we also use the root mean square error (RMSE), which is defined as Equation (Equation 14), where $\hat{y}$ and *y* denote the predicted and true depression levels, respectively, and *N* represents the number of samples.
(14)$$\begin{array}{c} \mathrm{RMSE}=\sqrt{\frac{\sum_{i}^{N}\left(y_{i}-{\hat{y}}_{i}\right)}{N}}. \end{array}$$

### 4.4. Experimental Setup

To demonstrate the effectiveness of our proposed method, we apply it, along with the original baseline GRU model [5], to obtain a direct comparison between them. The AVEC 2019 DDS dataset is split into training, development, and test sets. We utilized only the training and development sets for a fair comparison with the state-of-the-art methods following [3,4]. Our experiments were conducted on 219 subjects: 163 subjects for training and 56 subjects for development. The Adam optimization algorithm [22] was employed to learn the parameters in our networks. The learning rate was set to $1\times 10^{-5}$. The batch size was set to 48 for low- and middle-level features and 24 for high-level features. We trained our model for 500 epochs for low and middle-level features and 200 epochs for high-level features. During training, we set a,b to 1.0, 0.0 for single-task and 0.9, 0.1 for multi-task in the loss function of Equation (Equation 8). For the Transformer Encoder block, we set the head number of Multi-head Attention to 1, the hidden dimension of the Feed-Forward layer to 2048, and the number of the encoder layer to 6 following the original Transformer structure [8]. Our model is implemented with the framework of PyTorch [23], whereas our experiments are conducted on double Nvidia RTX 3090 GPU cards.

Our proposed networks have several hyperparameters to be optimized. The length of inputs for the Transformer Encoder (*N*) and the number of selected modalities for fusion (*M*) are the most important architectural decisions. After the exploration, we select *N = 2048* for low- and middle-level features and *N = 720* for for high-level features in this work. For *M*, M=3 and M=6 are the best choices for multi-task and single-task in Adaptive Late-Fusion, respectively. We will elaborate our exploration procedure in more detail in the Section 6.

## 5. Results

In this section, we will describe the results of our experiments. We first discuss the effect of the selection of the Transformer Encoder. Then, we describe the effectiveness of multi-task learning and multi-modal learning. Finally, we compare our results with those of some state-of-the-art methods.

### 5.1. GRU vs. Transformer Encoder

To investigate the effect of transformer-based networks on the CCC scores and RMSE values, we use single features as inputs of the networks, and the task is set to PHQ-8 regression. The results are presented in Table 2.

As presented in Table 2, the transformer model outperforms the GRU baseline model [5] in terms of the CCC metric for low- and high-level features. The CCC score of the AUpose feature is higher than those of other types of features. The transformer-based network achieves higher accuracy for low and high-level features, so we can conclude that the transformer model outperforms the GRU [5] in terms of processing low- and high-level features. However, for middle-level features, we can deduce that the transformer-based network does not outperform the GRU model.

### 5.2. Single-Task vs. Multi-Task

To determine whether the proposed model benefits from multi-task learning, we compare single-task results (PHQ-8 regression) with multi-task results (PHQ-8 regression and 5-class classification). As presented in Table 3, multi-task representation learning for PHQ-8 regression with 5-class classification for depression level detection exhibits better performance than single-task representation learning. The results indicate that depression detection can be improved using multi-task representation learning.

### 5.3. Single Modality vs. Averaged Multi-Modal Fusion vs. Adaptive Multi-Modal Fusion

To illustrate the effectiveness of multi-modal learning, our proposed method has been tested on all features available in the AVEC 2019 DDS dataset. As presented in Table 4, applying multi-modal late-fusion outperforms any uni-modal learning in any tasks, except for the Averaged All Late-Fusion. The Averaged All Late-Fusion has poor performance as it does not take the importance of different modalities into account.

The result indicates that the fusion of the best top *M* modalities improves the estimation of depression levels in terms of the CCC metric better than other fusions as modalities with poor performance are excluded to avoid a negative impact on the accuracy of depression detection. We can infer that the Adaptive Late-Fusion strategy can perform better than the Average Late-Fusion in estimating the levels of depression.

To investigate the different weights between different features, we counted the best three features with their corresponding weights in Adaptive Late-Fusion because M=3 is the best choice for multi-task learning and nearly the best choice for single-task learning. As presented in Table 5, although the selections of modalities are different for different tasks, the main influencing features are AUposes and MFCC_Functional. The results indicate that low-level features are more important than deep- and middle-level features for estimating the levels of depression.

### 5.4. Comparison with State-of-the-Art Methods

In Table 6, our approaches are compared with other state-of-the-art methods and the baseline. The baseline model [5] uses a GRU to extract temporal information and then takes the average of the results from every uni-modality. The hierarchical Bi-LSTM [2] hierarchically employs a Bi-LSTM to obtain temporal sequence information. Multi-scale temporal dilated CNN [7] employs dilated CNNs with different scales to process temporal information, followed by average pooling and max pooling to fuse temporal features. It should be noted that multi-scale temporal dilated CNN [7] uses features from texts extracted from pretrained models. Bert-CNN and Gated-CNN [14] use the Gated-CNN to extract features from each audiovisual modality sequence and the Bert-CNN to obtain features from texts before fusing the features to predict the final depression levels. The results indicate that our baseline has superior performance over other DL methods. The comparison of the predicted results with the ground truth is presented in Figure 4, and samples with different classification labels are colored differently.

## 6. Discussion

### 6.1. Effect of Frames Lengths

The length of inputs (i.e., the number of sequence frames) affects the accuracy of the depression detection of the networks and should thus be selected carefully. To study how the performance of our proposed method changes as we modify the length of input frames, we fix the task as regression and modalities as the fusion of top three features (*M* = 3) and compare the CCC score and RMSE at different selections of frames. As feature dimensions significantly differ, we select the same frames for low- and middle-level features, whereas we select different frames for high-level features. Although the use of the transformer model can capture long-term information, it consumes a lot of memory. The pair of *2048/720* frames for low and middle-level features and high-level features is the limitation of our hardware. As presented in Table 7, an increase in input frames improves the results. We select *N = 2048* for low- and middle-level features and *N = 720* for for high-level features in this work. Here *N* is the number of frames.

### 6.2. Effect of the Selection of Features

To study the effect of feature selection in the late-fusion, Figure 5 presents the results of different *M* selection in terms of the CCC scores with different multi-task combinations. The most appropriate number of selected top modalities for different tasks varies. The results indicate that the CCC scores increase with an increase in the number of top modalities selected and reach a maximum at M=3 for both single-task and multi-task in Averaged Late-Fusion. M=3 and M=6 are the best choices for multi-task and single-task in Adaptive Late-Fusion, respectively. Therefore, we select the corresponding best *M* for different tasks in different fusion strategies.

### 6.3. The Robustness from Adaptive Late-Fusion

Compared with Average Late-Fusion, Figure 5 shows that using Adaptive Late-Fusion not only achieves good results but also increases detection robustness, implying that the inclusion of low performance features has a slightly negative impact on the detection results.

### 6.4. Limitations of Our Methods

As shown in Figure 4, the predicted results of participants with high scores tend to be on the lower side. The reason is due to the imbalance of the training samples. As shown in Figure 3 (Section 4.1), the training set distribution is unbalanced, more samples have participants with low PHQ-8 scores, whereas few samples have participants with high scores. As a result, our model predicts a slightly lower PHQ-8 score than the true label for participants with high scores. The prediction accuracy can be improved by increasing the number of participants with high scores. It should be reminded that as data obtained from AVEC 2019 DDS have been processed to specific features for private issues, the data pre-processing stage in Figure 1 can be skipped. The pre-processing methods applied in the AVEC 2019 DDS Challenge dataset may affect the final performances for some detailed reasons, but it can provide the platform on which we can perform the fair comparison of modality processing and fusion. With current hardware, we think the pre-proceeding stage will have little limitation to the real-time detection scenario.

Although our methods have achieved good results, they still have many shortcomings. For instance, the depression level cannot be inferred from short-term sequences because it is a persistent long-term characteristic obtained from a human. Like many existing methods, the proposed method tends to pay more attention to the prediction accuracy by using long-term sequences for temporal feature extraction. How to extract effective features from short-term sequence and realize fast computer-aided diagnosis will be our future work.

## 7. Conclusions

In this study, we presented a multi-modal adaptive fusion transformer network for depression detection using multi-task representation learning with facial and acoustic features, which achieves the best results on the development set of the AVEC 2019 DDS dataset when comparing with other methods shown in Table 6. The experimental results indicated that the use of the transformer model for depression detection can improve the final prediction performance. Our ablation study demonstrated that multi-task representation learning, with tasks such as PHQ-8 regression and 5-class classification, outperforms single-task representation learning for depression detection. The experimental results indicated that Adaptive Late-Fusion contributes more significantly than Averaged Late-Fusion to the depression detection performance while also increasing robustness when fusing bad-performance features. By fusing the selected modalities, our proposed approach achieved a CCC score of 0.733 on the AVEC 2019 DDS dataset, outperforming the alternative methods investigated in this work.

## Figures and Tables

**Figure 1 sensors-21-04764-f001:**
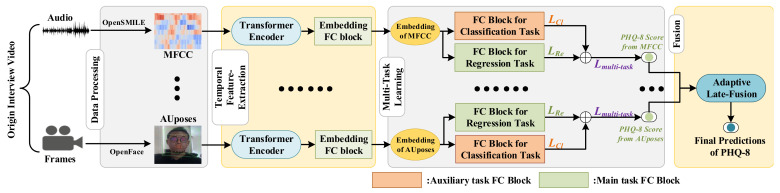
Overview of our proposed method. The origin data is firstly processed in the *Data Processing* Stage which has been done in the AVEC 2019 DDS dataset. Then the *Transformer Encoder* is used for extracting temporal information. The *Embedding FC Block* combined by a RELU activation layer, a dropout layer and a fully-connected layer is used to extract the hidden embeddings representing each kind of feature. The embeddings from each kind of feature are fed to two *FC blocks* to perform multi-task predictions. Finally, the results from different features are fused in the Adaptive Late-Fusion to predict the final results. The $L_{Re}$ means the Concordance Correlation Coefficient Loss for PHQ-8 regression and the $L_{Cl}$ means the Cross Entropy Loss for 5-class classification.

**Figure 2 sensors-21-04764-f002:**
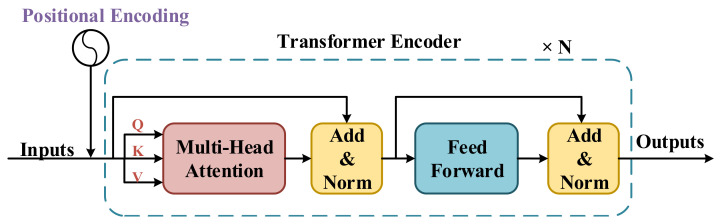
The structure of the Transformer Encoder employed to extract temporal information of sequences. After the data processed, the data are fed to the Transformer Encoder to extract temporal information. A single Transformer Encoder layer is composed by a Multi-Head Attention module and a Feed-Forward module with an external Positional Encoding module.

**Figure 3 sensors-21-04764-f003:**
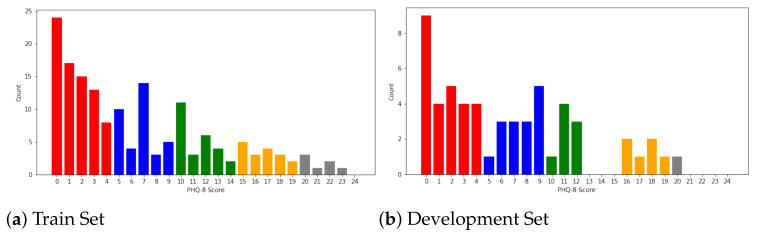
The distribution of the training and development set of the AVEC 2019 DDS Challenge dataset.

**Figure 4 sensors-21-04764-f004:**
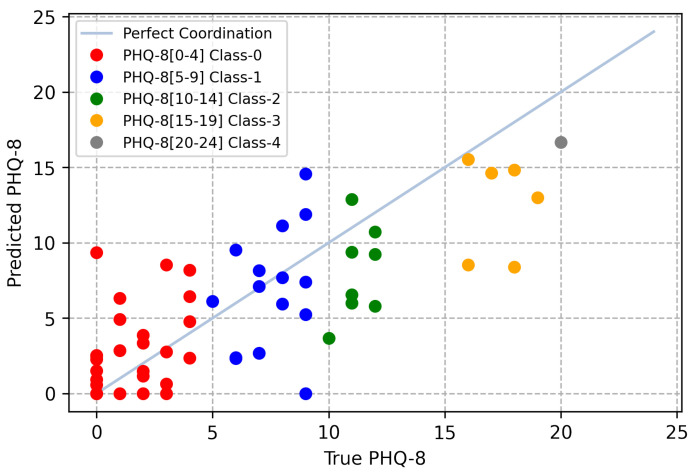
Correlation graph between the predicted and true PHQ-8 scores. Each color represents different classes.

**Figure 5 sensors-21-04764-f005:**
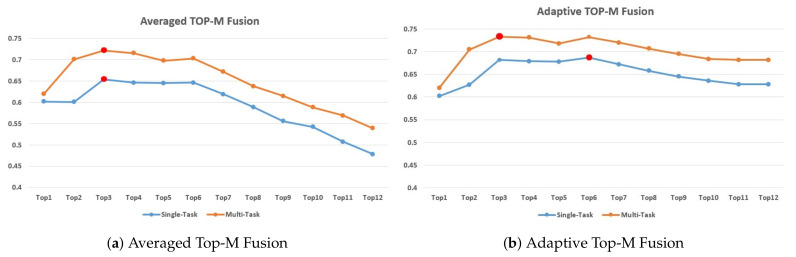
The CCC scores for different number of top M modalities fusion. Each color represents different tasks. The points with best results are marked as red.

**Table 1 sensors-21-04764-t001:** Distribution of Training and development splits with the relationships between 5-class classification labels and PHQ-8 regression labels.

Task		Train	Dev
Regression Task		163	56
Classification Task	minimal [0–4]	77	26
	mild [5–9]	36	15
	moderate [10–14]	26	8
	moderately severe [15–19]	17	6
	severe [20–24]	7	1

**Table 2 sensors-21-04764-t002:** Comparison between The GRU baseline model [5] and The Transformer-based model for different features from audio and video modality. For every kind of feature from each modality, we use two metrics including CCC and RMSE. The data in bold means the results with better performance.

Feature		CCC		RMSE	
		Baseline [5]	Proposed Method	Baseline [5]	Proposed Method
Low-level	MFCC	0.198	**0.289**	7.28	**5.70**
	MFCC_functional	-	**0.386**	-	**7.78**
	eGeMAPs	**0.076**	0.0002	**7.78**	8.69
	eGeMAPs_functional	-	**0.138**	-	**8.10**
	AUposes	0.115	**0.602**	7.02	**5.64**
	AUposes_functional	-	**0.277**	-	**6.23**
Middle-level	BoW-MFCC	**0.102**	0.060	**6.32**	8.58
	BoW-eGeMAPs	**0.272**	0.169	**6.43**	8.56
	BoW-AUposes	0.107	**0.210**	**5.99**	9.045
High-level	Deep Spectrogram_DNet	0.165	**0.204**	**8.09**	8.67
	Deep Spectrogram_VGG	**0.305**	0.141	**8.00**	8.72
	Facial_ResNet	0.269	**0.373**	7.72	**7.56**

**Table 3 sensors-21-04764-t003:** Comparison between single-task and multi-task with metrics of CCC, RMSE. Single-Task includes the PHQ-8 regression task while Multi-Task includes the PHQ-8 regression task and the 5-class classification task. The data in bold means the results with better performance.

Tasks	CCC	RMSE
Our: Single-Task	0.679	4.150
Our: Multi-Task	**0.733**	**3.783**

**Table 4 sensors-21-04764-t004:** Comparison between single modality and multiple modalities fusion and Comparison betweenn averaged multi-modal fusion and adaptive multi-modal fusion. We employ CCC and RMSE as metrics. For every kind of fusion strategy, we perform two ways of late fusion including *all fusion* and *top M fusion*. The data in bold means the results with better performance.

	CCC		RMSE	
	Single-Task	Multi-Task	Single-Task	Multi-Task
MFCC	0.289	0.471	5.700	5.158
MFCC_functional	0.386	0.460	7.780	7.269
eGeMAPs	0.0002	0.000	8.688	8.688
eGeMAPs_functional	0.138	0.107	8.102	8.345
AUposes	0.602	0.620	5.643	5.355
AUposes_functional	0.277	0.390	6.227	7.114
BoW_MFCC	0.060	0.063	8.584	11.575
BoW_eGeMAPs	0.169	0.181	8.560	8.555
BoW_AUposes	0.210	0.184	9.045	10.760
DeepSpectrogram_Dnet	0.204	0.171	8.672	8.662
DeepSpectrogram_VGG	0.141	0.170	8.721	8.145
Facial_ResNet	0.373	0.360	7.561	6.900
Averaged all fusion	0.478	0.539	4.591	4.334
Averaged best top M fusion	0.654	0.722	4.602	3.852
Adaptive all fusion	0.628	0.682	4.046	**3.782**
Adaptive best top M fusion	**0.687**	**0.733**	**3.829**	3.783

**Table 5 sensors-21-04764-t005:** The selection of modalities in TOP-3 Adaptive Late-Fusion and their corresponding weights.

Tasks	Best 3 Features and Corresponding Weights			
Single-Task	Modality	Video	Audio	
	Features	AUposes	MFCC	MFCC_Functional
	Weights	0.40	0.30	0.30
Multi-Task	Modality	Video		Audio
	Features	AUPoses	ResNet	MFCC_Functional
	Weights	0.44	0.27	0.29

**Table 6 sensors-21-04764-t006:** Comparison of the proposed method and the state-of-the-art with CCC metrics and modalities used. The data in bold means the results with better performance.

Methods	CCC	Modalities Used
Baseline [5]	0.336	Audio/Video
Hierarchical BiLSTM [2]	0.402	Audio/Video/Text
Multi-scale Temporal Dilated CNN [7]	0.466	Audio/Video/Text
Bert-CNN & Gated-CNN [14]	0.696	Audio/Video/Text
Ours Best	**0.733**	Audio/Video

**Table 7 sensors-21-04764-t007:** The CCC and RMSE results for different frames of input on fixed 3-top modalities fusion regression single task. *2048/720* means that we select *N = 2048* for low-level and middle-level features and *N = 720* for high-level features. The data in bold means the results with better performance.

Frames (*N*) (Low&Middle-Level Features/High-Level Features)	CCC	RMSE
2048/720	**0.654**	**4.602**
1536/540	0.634	4.526
1024/360	0.560	5.102

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
