# Peer review of "Multi-Modal Adaptive Fusion Transformer Network for the Estimation of Depression Level"

_sensors, 2021, doi:10.3390/s21144764_

Round 1
Reviewer 1 Report
The manuscript presents a multimodal transformer network-based technique to predict the level of depression. The objectives and results seem interesting, although several deficiencies in the organization of the work introduce confusion. These lacks impede a proper analysis of the impact of findings.
The following list of issues summarizes its major lacks.
- Abstract. Try to avoid citing of examples and the expresion "to the best of our knowledge" in this section. Please summarize the state of the art, justified objectives and succinct method, and present major conclusions. Authors are proposing three goals in the abstract, but the challenges addressed in the study are two (lines 51-52). This could be better clarified in Abstract. The public dataset used to validate the method is that pointed in the Abstract? Then the term "i.e.," is not necessary.
What is the mean of "state of the art performance" in this context?
What is the significance of 0.733 against 0.69 in the CCC in this study? It is not clear in this Section if those metrics refer to the same data population. They must clarify this issue.
2. Introduction. Is there any external reference that justifies "Long-term sequences has always been a challenge in DL"?. Moreover, long-term sequences have only a significance in the context of off-line procedures to detect depression, but a method designed for a sensor technology could be better under a real-time scenario. The authors do not address this issue.
Line 38- Sources for RNN in detecting depression.
Line 41. Sources for the "AVEC 2019 Detecting Depression with AI Sub-challenge (DDS)". Clarify the meaning of AI subchallenge.
Many blocks of text must be clarified. Example: Line 46 - "The forgetting issue of RNN means that ...", "Second, there have always been various fusion approaches to fuse information from different modalities".
Line 52. Clarify what two challenges the authors refer to in this sentence.
Line 60. Clarify what is a main versus minor modality.
3. Related work. The introduction could be improved if the related work is translated to that Section. The resulting Section could give a more comprehensive and completed Section that helps to justify the ultimate goal of the study.
4. Proposed Methods. Proposed techniques or algorithms?
Line 137 - "As data obtained from AVEC 2019 DDS have been processed to specific features, the data pre-processing stage in Figure 1 can be skipped": this issue needs to be commented in discussion. What type of pre-processing was applied previously to this study? How could this pre-processing process affect to the use of the presented technique on a real-time sensor platform? Is this a limitation?
How is coded the variable position, pos, in equation (1)?
Line 177 - formulaic?
The SoftMax () functional code in equation (2) must be clarified.
How the three linear transformations stream Q,K, V, are generated? Equation (2)
Define all the variables in Equation (4).
5. Experiments and Results. The experimental methodology could be separated from the Results to clarify the study.
Line 241 - "Because the data sequences are too long to fit in memory": This is a very vague claim. It depends on the target hardware to be used. The authors do not address this issue in the study.
Experimental setup. Clarify the selection of only training and development sets "for a fair comparison with the state-of-the-art method" (Line 271-).
4.5 Results. Line 284 - . It is confusing the sentence: "In this section, we will describe the results of our experiments in more detail." This is the beginning of the Results, and thus the results must be presented here and not before.
Authors combine results with the discussion of results. For example, they analyse the influence of the inputs’ length (4.5.1), without a previous design of this issue in the methods Section. This deficiency and others similar should be corrected.
6. Discussion. This section is too brief, due partly to describe some discussion issues in previous Sections (for instance the effect of frame lengths). The authors do not analyse the limitations of the methods (learning rate, etc.), data, and scenario. With respect to the scenario, are they considering that this technology could be implemented in a target sensorics hardware?
Figures 4 and 5 that are shown and commented in Discussion are Results indeed. These figures must be planned in a Methods Section, presented in a Results Section, and analysed in the Discussion Section.
7. Conclusion. Line 370 - "which achieves the best results on the development set of the AVEC 2019 DDS dataset". It must clarify that this performance occurs with respect to others compared techniques.
Other asseverations like (Line 375 - ) "However, the results indicated that the combination of the regression task and the binary gender classification task cannot outperform the combination of the regression task and the 5-class classification task" should be better addressed in the Discussion Section.
Author Response
Dear editors and anonymous expert reviewers,
Please see the attachment.
We greatly appreciate your great efforts of reviewing this submission and all the constructive comments, which have proven invaluable in helping us to improve our work. We have revised our paper based on reviewers’ comments. The revised parts are highlighted with Red in the revised paper. Detailed revision and answers to reviewers’ comments are summarized in the attachments.
Hao Sun

Reviewer 2 Report
A thorough proofreading is required. There is a lot of repetitions even in one line (e.g.: fuse, proposed method, forgetting, outperform), synonyms should be used.
Punctuation should be checked (e.g.: commas after formulae 1-8 and full stop after 9).
Captions under drawings and before tables should be short, the description of the results in the text. For Figure 3 caption, please, check the template once more.
The references in Section 2 should be introduced in a bit different manner, e.g.: Name A, ed al. in [2] used... In paper [4] method x has been proposed...In [8] the authors fused...
In few places (lines 178, 179, 267, 333) uppercase letters should be italicized .
In line 231 and table 3 caption spaces are missing.
For more detailed, please see the attached file.

Author Response
Dear anonymous expert reviewer,
Please see the attachment.
We greatly appreciate your great efforts of reviewing this submission and all the constructive comments, which have proven invaluable in helping us to improve our work. We have revised our paper based on reviewers’ comments. The revised parts are highlighted with Red in the revised paper. Detailed revision and answers to reviewers’ comments are summarized in the attachments.
Hao Sun
